# Bio-Capture of Solid Pollutants by Vegetation Canopy Cave in Shallow Water Flow

**DOI:** 10.3390/ijerph16234846

**Published:** 2019-12-02

**Authors:** Yanhong Li, Liquan Xie, Tsung-chow Su

**Affiliations:** 1State Key Laboratory of Ocean Engineering, School of Naval Architecture, Ocean and Civil Engineering, Shanghai Jiaotong University, Shanghai 200240, China; yyhli@sjtu.edu.cn; 2College of Civil Engineering, Tongji University, Shanghai 200092, China; 3Department of Ocean and Mechanical Engineering, Florida Atlantic University, 777 Glades Road, Boca Raton, FL 33431, USA; su@fau.edu

**Keywords:** biotechnique, foliage capture, shallow water flow, solid pollutants

## Abstract

Vegetation has already been acknowledged to have the ability to remove fine solid pollutants by retention and absorption, and is widely used in the biorestoration engineering of natural shallow water flow. Vegetation usually takes a long time to obtain the expected removal rate. Therefore, vegetation is not applicable for some urgent or pressing situations. In addition, in traditional biorestoration engineering, solid pollutants usually deposit in the soil of flow bed, which infiltrates into the far-field and accumulates in crops to threaten human health. Herein, we propose a new biotechnique of foliage capture by designing a cave on the top of a vegetation canopy, which is aimed to enhance the removal efficiency (i.e., achieve quick removal) and avoid the soil deposition of pollutants. The effectiveness and efficiency of this new design were validated by a set of indoor water flume experiments, with one flat canopy top configuration serving as the model of a traditional bioretention system and three cave configurations of differing aspect ratios. The results showed that compared with that of the flat canopy top, the total amount of foliage-captured solid particles for the three caved canopies increased by 3.8, 7.3, and 12.2 times. Further, we found that the foliage-capture efficiency depended on the aspect ratio of the canopy cave. The results revealed that the effectiveness of foliage capture and the enhanced efficiency were mainly from three hydrodynamic mechanisms: (i) as flow penetrated the cave boundary from the above-canopy region to the within-canopy region, it entrained solid pollutants to collide with the foliage and increased their fate of capture; (ii) the large eddy vortices of turbulence broke due to the increasing canopy resistance, which resulted in enhanced mixing dynamics for fine, suspended, solid pollutants to collide into foliage; and (iii) the flow shear along the cave boundary decreased, which provided a reduced lift force for solid pollutants to suspend or resuspend. Comparisons between the flat canopy and caved canopies of three aspect ratios showed that the design of the canopy cave is highly significant for capture efficiency.

## 1. Introduction

In shallow water flow, such as small streams, artificial open channels, and lakes or reservoirs, fine solid pollutants do harm to the ecosystem and human health directly [1,2]. Fine solid pollutants can increase the turbidity of water and interfere with nutrition transformation among water, vegetation, and soil [3]. In addition, these pollutants reduce water quality in lakes or reservoirs, which provide potable water to residents. Further, fine solid pollutants can infiltrate into far-soil and then accumulate in crops, thereby endangering humanity by causing diseases [4]. 

Vegetation or trees have been widely acknowledged to have a significant effect on the removal of pollutants (e.g., Tunçsiper [5]; Guo et al. [6]), and are therefore usually used as the main phytoremediation element in ecorestoration and bioretention engineering [7]. Elzein et al. [8] successfully used a constructed wetland to treat waste water to meet the requirements of reuse in urban communities. They found that vegetation played a very important role in purifying wastewater, and that the roots of vegetation provided a huge surface area for attaching microbial growth. Tunçsiper [5] designed a natural waste water treatment system which included a settlement basin, a free water surface flow constructed wetland planted with *Phragmites communis*, and an overland flow system planted with Italian ryegrass. The system removed 85% of the total suspended solids and up to 49% of phosphorous. Alemu et al. [9] investigated the effect of herbaceous buffer strips on the reduction of sediment in the agricultural season of 2015. They reported that permanently vegetated areas located between the crop field and a stream in the East African highlands resulted in the filtration of 94% of the total suspended sediment.

The markable effect of pollutant removal is mainly achieved by the physical deposition and filtration of solid particles by the roots of vegetation. Abdelhakeem et al. [10] investigated the removal rate of total suspended sediment in an artificial wetland constructed from plastic and weeds with the dimensions 0.3 m × 0.3 m × 0.3 m. During the eight-month study, 75% of the total suspended sediment was deposited and filtered due to the reduced flow velocity, reinforced settling velocity of the sediment, and the enhanced filtration process through the extensive root system of the reeds. Leguédois et al. [11] investigated sediment trapping by a tree belt in an overland flow. They concluded that the main sediment trapping process (62% of the total trapped sediment) occurred in the hydraulically changed flow zone, i.e., the backwater zone. Ünlü and Alpar [12] investigated the biofiltration system at a field scale from December 2006 to May 2007. They found up to 90% removal of suspended solids and heavy metals due to the developed root system of vegetation. 

There are two main shortages of the normal vegetation-involved and controlled restoration or bioretention system. The first one is that the above hydraulic settling or root-filtration methods usually take a long time (several months to years) to obtain the expected effect [9,12]. In some emergent circumstances, such as a concentrated leakage of pollutants from a factory to a stream, the normal hydraulic settling and vegetation-root-fixing methods no longer meet the urgent restoration requirements of an urban environment. In addition, although removed from the water column, the hydraulically settled and vegetation-root-fixed solid pollutants still stay in the bed-soil, and may stay for a long time—up to decades—before they are biodecomposed in soil or uptaken into leaves for bio-absorption [12]. These deposited pollutants can infiltrate far into the crop fields to endanger humanity directly. Consequently, the improvement of removal efficiency and avoidance of excessive soil retention of solid pollutants are two challenges in ecorestoration and bioretention engineering in shallow water flow. 

The objectives of this paper are to propose a foliage-capture technique for the removal of solid pollutants with high efficiency, and to validate this technique’s effectiveness and efficiency. The working mechanism of this new biotechnique is hydrodynamic. The physical process of foliage capture involves: (i) the design of a cave atop a submerged vegetation canopy, (ii) the induction of large-scale secondary flow due to this cave (i.e., an apparent curve of streamline), (iii) the modification of local flow velocities due to the secondary flow and interference with the evolution of turbulent vortices, (iv) the combination effect of reduced flow velocity and broken vortices resulting in an increase in the collision probability between suspended solid pollutants and vegetation foliage, and (v) finally, the quick capture of suspended solid pollutants by vegetation foliage. 

## 2. Experimental Set-Up and Methodology

### 2.1. Water Flume and Coordinate System

A set of experiments were conducted in an indoor straight water flume with a rectangular cross-section. The dimensions of the flume were 12 m long, 40 cm wide, and 40 cm high (Figure 1 and Figure 2). The coordinate system is shown in Figure 1 and Figure 2, with the origin located at the cross-point of upstream edge and the streamwise center line of flume bottom. The two side-walls of the water flume were made of transparent glass, which enabled outside observation of the topography configurations of the submerged vegetation canopy and assisted in the inside installation of experimental devices. A water tank of dimensions 1 m × 1 m × 0.5 m was set 1.0 m above the flume at the upstream end (Figure 2). This water tank was connected to the water flume with an upper surface open corridor. The connecting corridor was 38 cm wide, 40 cm high, and has a 35° tilt with respect to the horizontal line. The elevation fall head of 1.0 m between the tank and the flume bottom provided energy to drive the experimental flow. Water depth in the tank was maintained at 0.3 m during each entire experimental set. A triangular wire was set at the downstream section of the flume to adjust flow depth (H) and measure flow discharge (Q) (Figure 2). 

### 2.2. Flow Conditions and Vegetation Canopy Configurations

Note: The flat canopy top configuration was set as the control test to model that in traditional bioretention or biorestoration systems.

The bed slope of the flume was set at 0.0022 to keep the experimental flow in a quasi-steady and uniform state. During the total four sets of experiment in the present study, the flow discharges (Q) were kept at a constant value of 8100 cm^3^/s and the flow depth (H) was 23 cm (Table 1). 

The vegetation was modeled with polyamide plates to enable fine suspended solid pollutants to adhere. This material is rigid and prevents vegetation from deforming or swaying in flow. The vegetation foliage consisted of a plate with dimensions 7.5 cm high, 1.2 cm wide, and 1 mm thick (Figure 3a). The vegetation foliage was installed on the flume bed in a staggered distribution pattern (Figure 3b). The vegetation canopy occupied the flume bed from x=1.1 m (at the downstream side of the particle injection board) to the downstream end of the flume. 

In nature, aquatic vegetation usually grows in variable distribution densities. Where the distribution density is sparse, the submerged vegetation canopy is easily depressed by the downward flow fluctuation, thereby forming caves (i.e., small pits) in its topography. To model the sparse-canopy-induced canopy cave, a two-dimensional depressed cave on the top of the vegetation canopy was designed in the middle section of the flume in the x−z plane, centered at x=5 m from the upstream end of the flume (Figure 1). In the upstream and downstream sides of the canopy cave, i.e., in the undepressed (erect) canopy sections, the spacing between two neighboring vegetation stems was $S_{e}=$ 1.5 cm. In the depressed vegetation section, the spacing was sparser: *S_d_* = 3 cm (Figure 3b and Figure 4). The cave topography of the vegetation canopy was produced by increasing pre-bending of the rigid vegetation stems from the upstream and downstream edges of the cave towards its center part, thereby forming a concave curve shape (Figure 3b).

In total, four sets of experiments, including one set in a flat canopy top configuration (i.e., without a cave) and three sets with cave configurations, were performed. The two dimensions and aspect ratios of the four sets of canopy caves are given in Table 1. The without-cave configuration was set up as a control to model traditional bioretention or biorestoration systems, such as wet lands. 

Besides these four experimental sets, an additional experiment (set_0) was applied to measure the fine, suspended, solid pollutant concentration profile at x=3.5 m (Figure 2), which was conducted under the same flow and without the cave canopy condition as set_1. The solid pollutant samples were taken by a sampling bottle. Three sampling runs were conducted with the first run starting from 5 min, the second run from 35 min, and the third run from 65 min after the flow surface line became steady. In each sampling run, three rigid rods with four sampling bottles attached to each rod were applied (Figure 5). By adjusting the elevations of these four sampling bottles in four sampling runs, a total of 16 sampling points were obtained for each vertical profile. 

### 2.3. Fine Solid Pollutants and Observation Method 

Fine, suspended, solid pollutants were modeled using sphere glass balls of diameter 20 μm. These balls were injected into the experimental flow from nine evenly distributed points on a releasing board (Figure 6), which was installed at the upstream end of the flume. After the experimental flow became steady, the fine solid pollutants were injected at a speed of 0.9 g/s from each injection point. Thus, the releasing speed was 8.1 g/s on the entire cross-sectional area of the flume. The releasing process for the solid pollutants lasted for 30 min in each experimental set, and the experimental flow was stopped 5 min later. After each test set, the experimental water, including undeposited fine solid pollutants, was drained from three holes on the bottom of the flume at the downstream end. The vegetation stems in the range of the canopy cave (Figure 4) were then cut down from their bases and dried in an oven. The quantity of cave-captured fine solid pollutants on the vegetation canopy in the cave section was measured from comparison of the weight of the vegetation before and after the experiment. For the purpose of comparison, cave capture in downslope and upslope sections was measured independently and separately.

### 2.4. Flow Field Observation

Flow velocity field and the streamline around the depressed canopy cave were observed by hydrogen-bubble-based PIV (Particle Image Velocimetry) technique. By this method, hydrogen bubbles were used as the flow-tracing solid pollutants. The hydrogen bubble generating system was composed of a platinum wire with diameter 25 μm and an automatic synchronized power controller (ASPC). The platinum wire was vertically installed at the upstream edge of the depressed canopy cave (Figure 7). The bottom and top ends of the wire were connected to the flume bed and the negative electrode of the ASPC, respectively. The platinum wire was strained straight. The positive electrode of the ASPC was connected to a copper plate which was immerged in an isolated water tank. During the experiments, the ASPC electrolyzed the water adhering to the platinum wire at a low voltage of 6 V and a frequency of 60 Hz. Thus, it generated hydrogen bubbles at a frequency of 60 Hz along its underwater part. These hydrogen bubbles, with an original diameter of 25 μm, tracked the streamlines throughout the full length of the depressed canopy cave. 

The traces of the hydrogen bubble trackers were recorded with a set of image-recording systems composed of a laser light emitter and a video camera. The laser light emitter was hung at 45 cm above the water surface over the depressed canopy cave (Figure 2). A crystal cylinder of diameter 0.4 cm was installed beneath the laser light emitter to refract the laser light beam into a piece of laser light sheet in the x−z plane. The laser light sheet was aligned with the longitudinal centerline of the flume through adjusting the hanger. The hydrogen bubbles in the flow reflected the laser light. A video camera outside the flume was set up normal to the laser light sheet to record the traces of the hydrogen bubbles. The shooting frequency of the video was set at 120 snapshots per second. Thus, the flow velocity field and the streamlines were obtained by analyzing the successive video snapshots with the window-based range interrogation algorithm [13,14]. The hydrogen-bubble-based PIV technique was validated by the traditional PIV technique using glass ball particles as flow tracers.

In this hydrogen bubble technique, the diameter of the hydrogen bubble was 25 μm (the same as the diameter of the platinum wire) and the hydrogen bubbles in one vertical line were consistent. In the post-process of obtaining time series of flow velocity, the space between two vertical points was chosen to be 100 μm. Therefore, the measurement points of the time series of flow velocity were consistent and the vorticity can be calculated based on the measured data. 

### 2.5. Calculation of Vorticity and Shear Velocity

For the two-dimensional flow in the x−z plane, the second-order normal turbulence momentums $\bar{u^{\prime }u^{\prime }}$ and $\bar{w^{\prime }w^{\prime }}\;$(usually used to identify turbulence strength or intensity, $\sqrt{\bar{u^{\prime }u^{\prime }}}$ and $\sqrt{\bar{w^{\prime }w^{\prime }}}$, e.g., Kolerski and Wielgat, 2014 [15]) are the two main components of turbulence kinetic energy which provide the fundamental mechanism for fine solid pollutants to suspend or resuspend. The second-order shear turbulent momentum ($-\bar{u\prime w\prime }$) can account for the mixing dynamics of fine solid pollutants to suspend or resuspend, where $u^{\prime }\;\mathrm{and}\;w\prime$ are the turbulence perturbations of flow velocities u and w in streamline and vertical directions, respectively; and top bar (“$\bar{}$”) over the variables denotes the time-averaged value.

The vorticity influences the suspension, resuspension, or settling of fine solid pollutants, which is defined as:(1)$$\omega =\frac{\partial w}{\partial x}-\frac{\partial u}{\partial z}$$

The shear velocity ($u_{*}$) is used to normalize flow velocity (u) and the turbulence momentums $\bar{u^{\prime }u^{\prime }}$, $\bar{w^{\prime }w^{\prime },}$ and $-\bar{u\prime w\prime }$. In submerged vegetated flow, the main resistance is mainly caused by the block effect of vegetation canopy instead of bed and wall friction. In addition, the maximum flow shear occurs near the canopy top between the within- and above-canopy flow layers. Therefore $u_{*}$ is calcuted from the maximum value of $-\bar{u\prime w\prime }$ at the canopy top region as (e.g., Ghisalberti and Nepf [16]):(2)$$u_{*}=\sqrt{{\left(-\bar{u\prime w\prime }\right)}_{max}}$$

## 3. Result

### 3.1. Vertical Profile of Fine Solid Pollutants in Front of the Caves

The vertical profile of the concentration of fine solid pollutants was observed in front of the canopy cave at x=3 m for experimental set 1 (Figure 8). Figure 8 shows that the concentrations of fine pollutant solids reached a maximum value near the canopy top, and then decreased toward the water surface and downward toward the flow bed. Therefore, the movement of these intensely concentrated solid pollutants near the canopy top region was important to the foliage-capture behavior in the depressed canopy cave.

The distribution profile of suspended solid particles in submerged vegetated flow depends on different flow, particle, and vegetation conditions. In the over-canopy region, it has been reported that the concentration of suspended particles mono-decrease from canopy top region toward water surface (e.g., Li et al. [17]; Wang et al. [18]). In the within-canopy region, Li et al. [17] reported an increasing trend of suspended sediment from the canopy top region toward the flow bed; however, the present study showed an inverse trend. The main reasons for this contradiction were that the denser canopy in the present study acted as an obstacle to both the flow and injected particles at the upstream section, and the more accelerated flow in the upstream section entrained more suspended particles upward to the canopy top region.

Although the available data of the vertical profile of suspended solid pollutants in submerged vegetated shallow water flow is strictly limited, a data set showed that a peak concentration value of mass (dye) occurred near the vegetation canopy [19]. The main reason for this phenomenon was that the flow in front of the vegetation canopy was diverted to the canopy top region due to the blocking effect of the canopy patch. As a result, the fine particles were carried upward to the canopy top region with the upward accelerating flow. 

### 3.2. Total Amount of Foliage Captured Solid Pollutants in the Canopy Cave Regions

The total amount of solid pollutants captured by the vegetation cave in the four experiments is given in Table 2. Compared with that for the flat canopy top configuration (set_1), the total amount of captured solid pollutants for the with-cave configurations (set_2, set_3, and set_4) was significantly higher. The total amount of captured solid pollutants increased with decreasing aspect ratios of the canopy caves. Compared with that of experimental set_1 for the flat canopy topography, the total amount of foliage-captured solid pollutants in experiment set 2 of the largest cave topography increased by 3.8 times. The amount of foliage-captured solid pollutants was further increased by 7.3 times in set_3, and then by 12.2 times in the smallest aspect ratio test of set_4. This indicated that the new biotechnique of using a canopy cave for foliage capture of fine solid pollutants was effective and highly efficient.

Our technique’s effectiveness and high efficiency could be attributed to the hydrodynamic mechanisms changed by the canopy caves, namely, the flow field, shear turbulent momentum, normal turbulent momentums (also used to identify turbulent intensities), and evolution process of turbulent vorticities. These mechanisms are explained in detail in Section 3.3.

### 3.3. Hydrodynamic Mechanisms of Foliage Capture of Fine Solid Pollutants

#### 3.3.1. Flow Velocity Field around the Canopy Caves

The curve, secondary flow induced by the canopy caves determine their entrainment effect on the fine, suspended, solid pollutants. The features of the cave secondary flow in the x−z plane can be examined through the two-dimensional flow field in vertical plane. The flow fields around the depressed canopy caves for experimental sets 1–4 were observed. To avoid the interference between the glass-ball-modeled fine solid pollutants and the hydrogen bubbles, the flow fields were observed under clear water conditions for the four different canopy cave configurations. 

Figure 9a–d shows the flow fields around the canopy caves for experimental sets 1–4. The flow velocity was time-averaged over 3 min. For the flat canopy top configuration (i.e., without a cave) in experimental set_1 (Figure 9a), all velocity vectors were uniform in the x direction; that is, the flow was straight from upstream to downstream; however, two different flow layers within and above the vegetation canopy were remarkable, which produced substantial upward shear stress to produce a lift force for fine solid pollutants to suspend or resuspend in a water column, thereby maintaining the highly concentrated, fine, solid pollutants (Figure 8) at this height level and reducing their potential of captive fate by vegetation foliage. 

For the with-cave configurations of three different aspect ratios in experimental sets 2–4 (Figure 9b–d), the flow fields presented some general characteristics. The directions of velocity vectors tend to curve along the cave boundaries, forming the cave secondary flow. In the downslope section of the canopy cave, the flow from the under-boundary layer penetrates the permeable cave boundary directly, joining with the high-speed flow rushing down from the above-boundary layer. Therefore, the magnitude of the velocity adjacent to the above-boundary of the canopy cave increases due to the mixing of these two flows. This increased velocity above the cave boundary can carry the highly concentrated, fine, solid pollutants near the canopy top with flow instead of dropping them off on vegetation foliage. In addition, the difference in velocities between the under- and above-canopy boundary layers increases. Consequently, the flow shear and lift force along the cave boundary will increase, thereby enhancing the possibility for fine solid pollutants to suspend or resuspend. As a result, the possibility for fine suspended solid pollutants to be captured by vegetation foliage at the downslope section decreases significantly.

In the upslope section of the canopy caves (Figure 9b–d), flow velocity along the cave boundary decreased significantly due to the blocking effect and increasing resistance of the higher and higher vegetation canopies. Thus, the velocity magnitude along the upslope cave boundary in the concave region was smaller than that in the convex region, which produced reverse flow shear and a negative lift force normal to the cave boundary (indicated by the red arrows in Figure 9b–d). The above features became more remarkable as the aspect ratios of the canopy caves increased. The negative lift force can drive the highly concentrated solid pollutants carried by the cave secondary flow just above canopy boundary in the concave side to move toward the convex side, thus increasing the collision probability between solid pollutants and vegetation foliage and promoting the capture probability of solid pollutants. 

#### 3.3.2. Streamlines around the Canopy Caves 

Figure 10a–d shows the streamlines and the evolution of vorticities around the canopy caves for the four experimental sets. Figure 10a shows the streamlines for the flat topography of the vegetation canopy. In correspondence with the flow field, the streamlines were straight and parallel in the longitudinal direction of the flume. Therefore, the probability of collision between vegetation foliage and the highly concentrated solid pollutants above the canopy top was small. 

Figure 10b–d shows that for all the three cave configurations, the streamlines can be divided into two sections: the downslope and upslope sections. In the downslope section, the streamlines tended to rise upward when approaching the cave boundary, and in the upslope section, the streamlines tended to rise again when departing away from the cave boundary. Above the cave boundary, the streamlines tended to surf along the longitudinal direction of the water flume. The surf of the streamline was caused by the rising water surface above the canopy cave. The turbulent flow in the downslope section was enhanced by the cave secondary flow and the penetration of flow from the under-cave boundary to the above-cave boundary. In the upslope section, the turbulent flow was enhanced by the breaking of the turbulence vortex. The enhanced turbulence around the canopy caves resulted in increased flow resistance and thus decreased bulk flow velocity, which led to surfs in both the water surface and streamlines. 

#### 3.3.3. Evolution of Vorticities in the Canopy Caves

Figure 10a–d also shows the vorticities of turbulent flow in the canopy caves. For the flat topography of the canopy configuration in experimental set_1 (Figure 10a), the vorticities were horizontally stratified, indicating that the coherent turbulence moved along the streamwise direction without any transformation. The coherent structure of turbulence was induced by the velocity difference between above- and within-canopy layers of the flow, which had been acknowledged as the mixing layer formed large eddies [16].

Figure 10b–d shows that for the three cave canopy configurations, the large eddies of the mixing layer transformed and stretched in the downslope section, but broke in the upslope section. The transformation and stretch in the shape and intensity of large eddies occurred in the horizontal and vertical directions simultaneously. For the largest aspect ratio of the canopy cave configuration (set_2, Figure 10b), the vortices stretched for the largest length in the streamline direction in the downslope section. In the upslope section, however, the broken vortices covered the shortest region, and the vorticities were of the smallest intensity. For the smallest aspect ratio of the canopy cave configuration (set_4, Figure 10d), the vortices stretched for the smallest length in the streamline direction in the downslope direction. In the upslope section, the broken vortices covered the longest region, and the vorticities were of the greatest intensity. For the medium aspect ratio (set_4, Figure 10c), the lengths of the stretched and broken vortices and the intensity of the broken vortices lay between the other experimental sets. 

For the three cave canopy configurations (Figure 10b–d), the stretched vortices in the downslope section developed along the cave boundary, causing the core zone of the vortices to be along the cave boundary as well. Consequently, the most concentrated, fine, suspended, solid pollutants can be entrained in the core zone of stretched vortices and moved with vortices together.

In the upslope section, the broken vortices resulted in sharply decreased vorticities along the cave boundary, which caused the fine solid pollutants to no longer be entrained by the flow. In addition, the flow tended to penetrate the permeable cave boundary from the concave side into the convex side (Figure 9), thereby carrying the fine solid pollutants of mass to collide with vegetation foliage. As a result, the captive possibility of fine solid pollutants by the vegetation cave was increased. 

#### 3.3.4. Normal and Shear Turbulent Momentums around the Canopy Caves

Figure 11 shows the profiles of normalized, second-order, normal turbulent momentum ($\bar{u\prime u\prime }/u_{*}^{2}$, $\bar{w\prime w\prime }/u_{*}^{2}$) and shear turbulent momentum ($-\bar{u\prime w\prime }/u_{*}^{2}$) around the canopy cave for the flat topography of canopy configuration in experimental set_1. At the canopy top, all of these three variables reached their maximum values with $\bar{u\prime u\prime }/u_{*}^{2}$ reaching 0.826, $\bar{w\prime w\prime }/u_{*}^{2}$ reaching 0.446, and $-\bar{u\prime w\prime }/u_{*}^{2}$ reaching 2.1. From the canopy top, upward and toward the water surface and downward and toward the flow bed, all these three variables decreased gradually. This was consistent with the general characteristics of regular (without-cave) vegetation-canopy-occupied flow (e.g., Nezu and Sanjou [20]). 

Figure 12a–c shows the profiles of $\bar{u\prime u\prime }/u_{*}^{2}$ and $\bar{w\prime w\prime }/u_{*}^{2}$ of experimental sets 2–4 for the three canopy cave configurations. Each subfigure shows five vertical profiles which present the evolution process of turbulence intensities along the streamwise direction. These five profiles were set to align with the upstream edge of the canopy cave, the middle point of the downslope section, the bottom-most point of the canopy cave, the middle section of the upslope section, and the downstream edge of the canopy cave. For all of these five profiles, the peak points of $\bar{u\prime u\prime }/u_{*}^{2}$ and $\bar{w\prime w\prime }/u_{*}^{2}$ occurred just above the cave boundary. This feature implied that the canopy-induced turbulent flow would provide a strong diffusion mechanism for fine solid pollutants to suspend above the cave boundary. In the downslope section, the stretched vortices can entrain fine, suspended, solid pollutants moving forward along the cave boundary; however, in the upslope section, the broken vortices can cause the highly concentrated, pollutant, solid particles to collide into vegetation foliage and be captured.

Figure 13a–c shows the profiles of $-\bar{u\prime w\prime }/u_{*}^{2}$ for the three cave canopy configurations. Much as with the strongest turbulence intensities, the peak points of $-\bar{u\prime w\prime }/u_{*}^{2}$ occurred just above the cave boundary. This feature indicated a strong mixing mechanism of turbulent momentum along the peak points’ connecting line, which increased the collision possibility between the fine, suspended, solid pollutants and vegetation foliage in the upslope section of the canopy cave. 

## 4. Discussions

Although there have been numerous of studies on bed deposition or suspension of fine solid pollutants in vegetated shallow water flow [21,22,23,24,25,26,27,28,29,30], cave capture behavior of fine suspended solids is rarely reported [22,31]. It is commonly observed that the suspended solid pollutants in flow are left on the submersed foliage of aquatic vegetation after a flood, especially on the top part when the vegetation canopy is depressed down to form a cave. Detailed descriptions and the underlying mechanisms of this phenomenon are still lacking. The observed amount of foliage-captured solid particles in the present study is a composite of the limited data. 

The flow field, turbulent momentum, and the evolution process of turbulent vortices depend on different conditions of flow, particle, and vegetation. Therefore, in practical applications, the optimal configuration of the canopy cave needs to be particularly designed based on the specified flow, solid particles, and vegetation conditions. 

## 5. Conclusions

A foliage-capture biotechnique for the removal of fine solid pollutants in shallow water flow was proposed. The effectiveness and efficiency were validated with physical indoor experiments. Compared with the flat canopy top topography, which modeled the usually used configuration in traditional bioretention or biorestoration systems such as wetlands, the newly proposed with-cave canopy configuration enhanced the total amount of foliage-captured solid pollutants by up to 12.2 times. Under the newly proposed with-cave condition, the pollutant capture efficiency depended on the aspect ratio of the canopy cave.

The hydrodynamic mechanisms of foliage capture in the canopy cave region were revealed: as flow in the upslope section of the canopy cave penetrated the cave boundary from the above-canopy region to the within-canopy region, it entrained solid particles allowing them to collide with foliage, and increased the probability of capture; the large eddy vortices of turbulence broke due to the increasing canopy resistance in the upslope section, which resulted in enhanced mixing dynamics for fine, suspended, solid pollutants to collide into foliage; and the flow shear along the cave boundary in the upslope section decreased, which provided a reduced lift force for the suspension of solid pollutants. 

## Figures and Tables

**Figure 1 ijerph-16-04846-f001:**
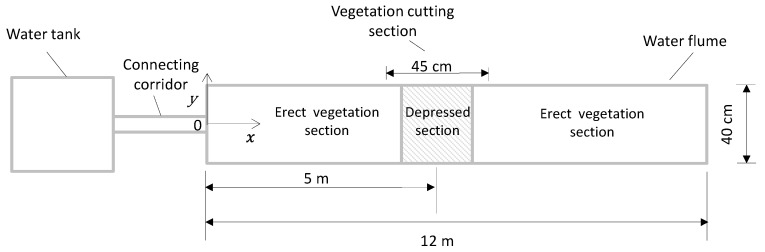
Horizontal planar view of the experimental set-up (not in scale).

**Figure 2 ijerph-16-04846-f002:**
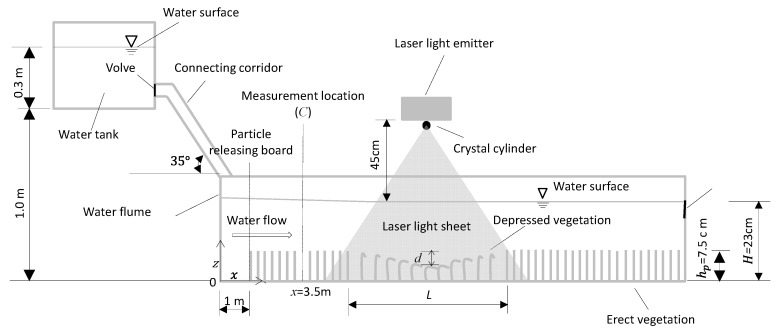
Side view of the experimental set-up (not in scale).

**Figure 3 ijerph-16-04846-f003:**
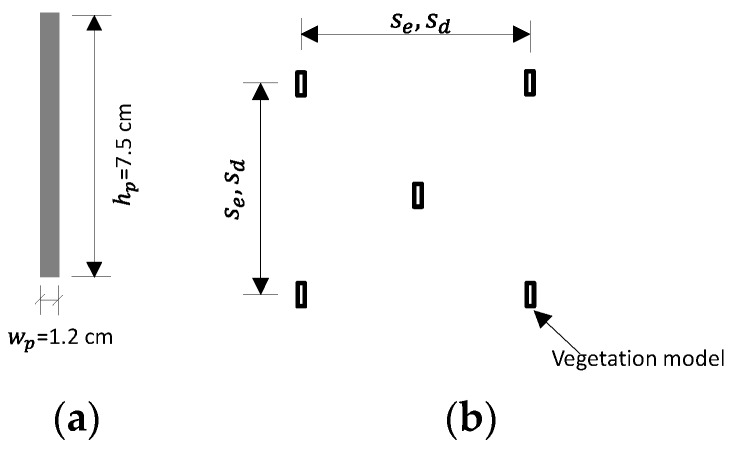
(**a**) Schematic of the vegetation model; (**b**) Staggered arrangement of the vegetation model.

**Figure 4 ijerph-16-04846-f004:**
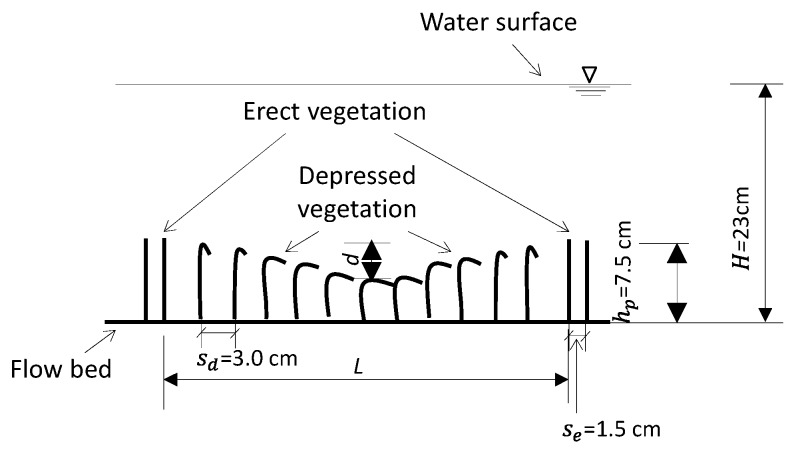
Schematic of the vegetation canopy caves.

**Figure 5 ijerph-16-04846-f005:**
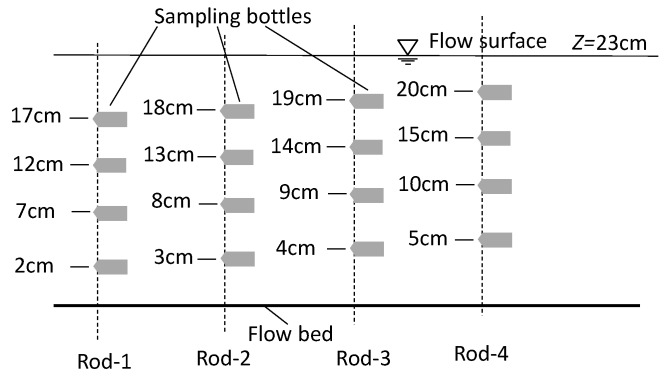
Elevations of sampling bottles.

**Figure 6 ijerph-16-04846-f006:**
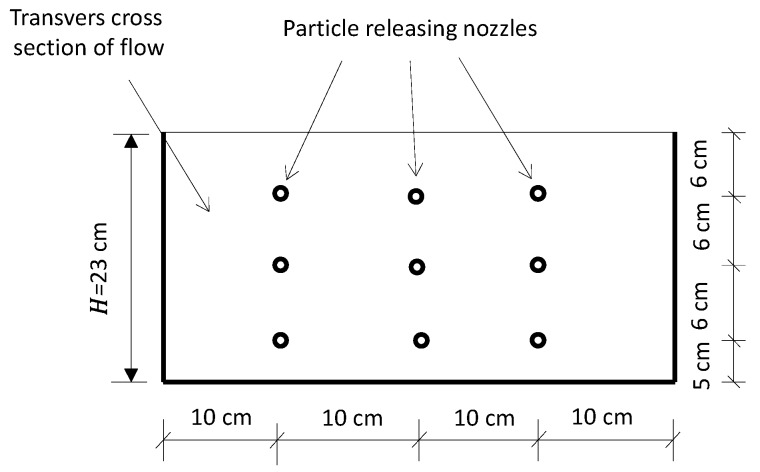
Solid pollutant injection points.

**Figure 7 ijerph-16-04846-f007:**
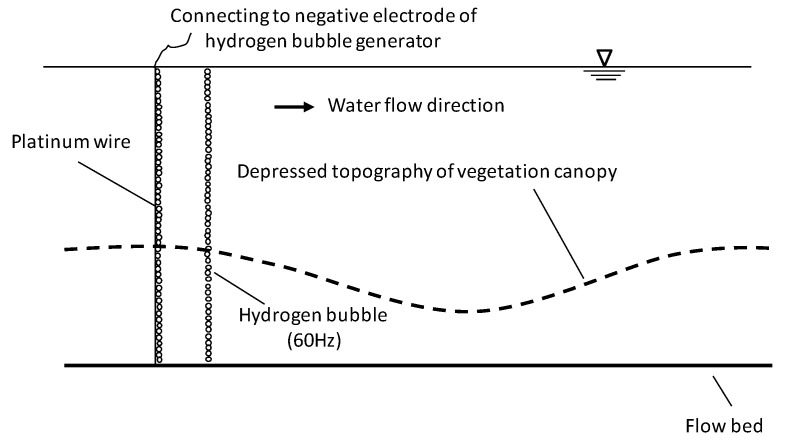
Schematic of the hydrogen-bubble-releasing system.

**Figure 8 ijerph-16-04846-f008:**
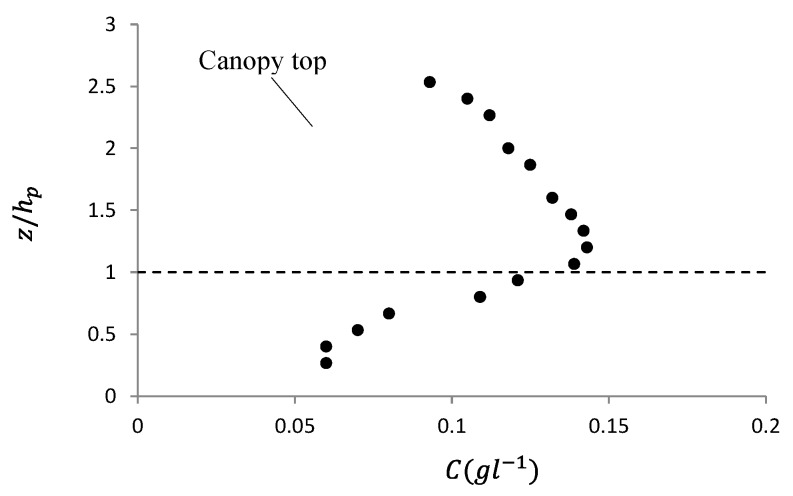
Vertical profile of suspended, fine, solid pollutants before the canopy cave at x=3 m.

**Figure 9 ijerph-16-04846-f009:**
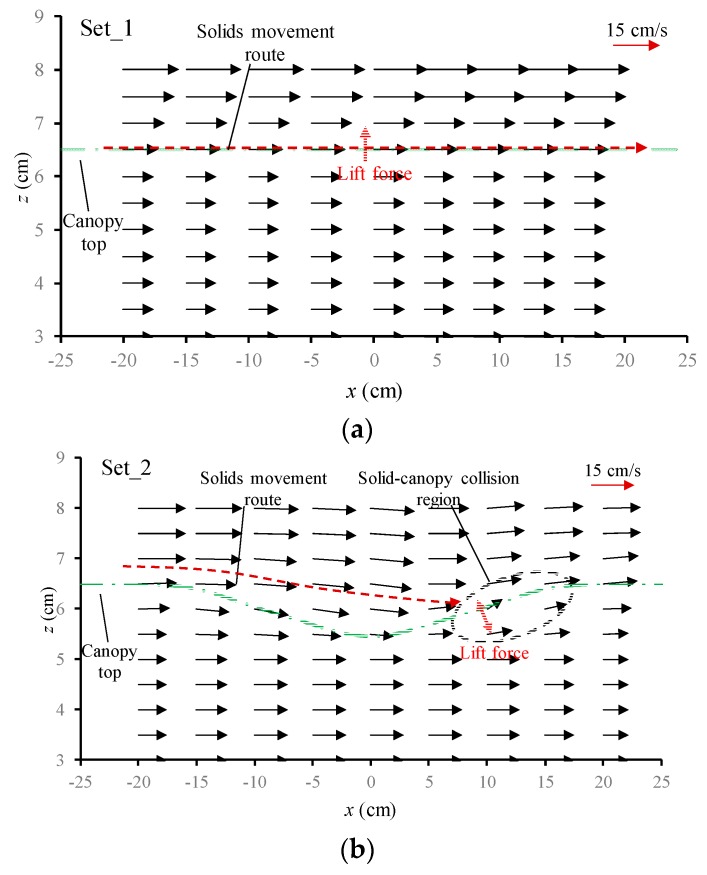
Velocity field for set_1 (**a**), set_2 (**b**), set_3 (**c**), set_4 (**d**).

**Figure 10 ijerph-16-04846-f010:**
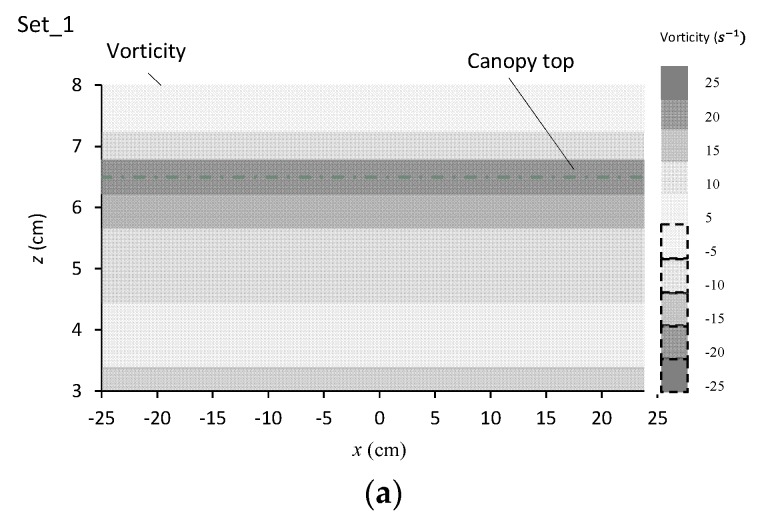
(**a**) Streamlines and vorticities around the canopy cave for set_1 (**a**), set_2 (**b**), set_3 (**c**), set_4 (**d**).

**Figure 11 ijerph-16-04846-f011:**
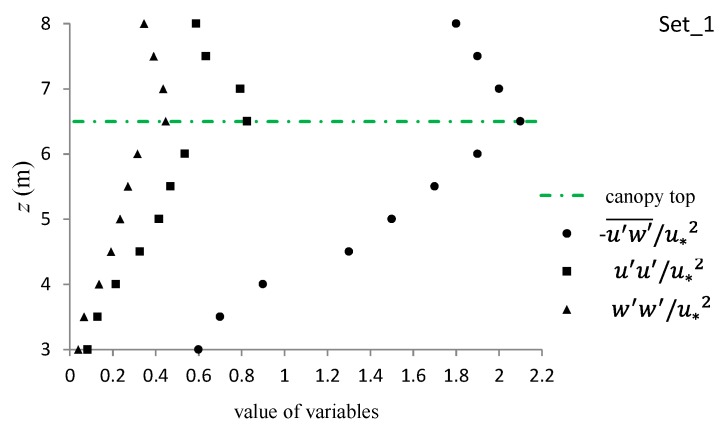
Profiles of $\bar{u\prime u\prime }/u_{*}^{2}$, $\bar{w\prime w\prime }/u_{*}^{2}$, and $-\bar{u\prime w\prime }/u_{*}^{2}$ for set_1.

**Figure 12 ijerph-16-04846-f012:**
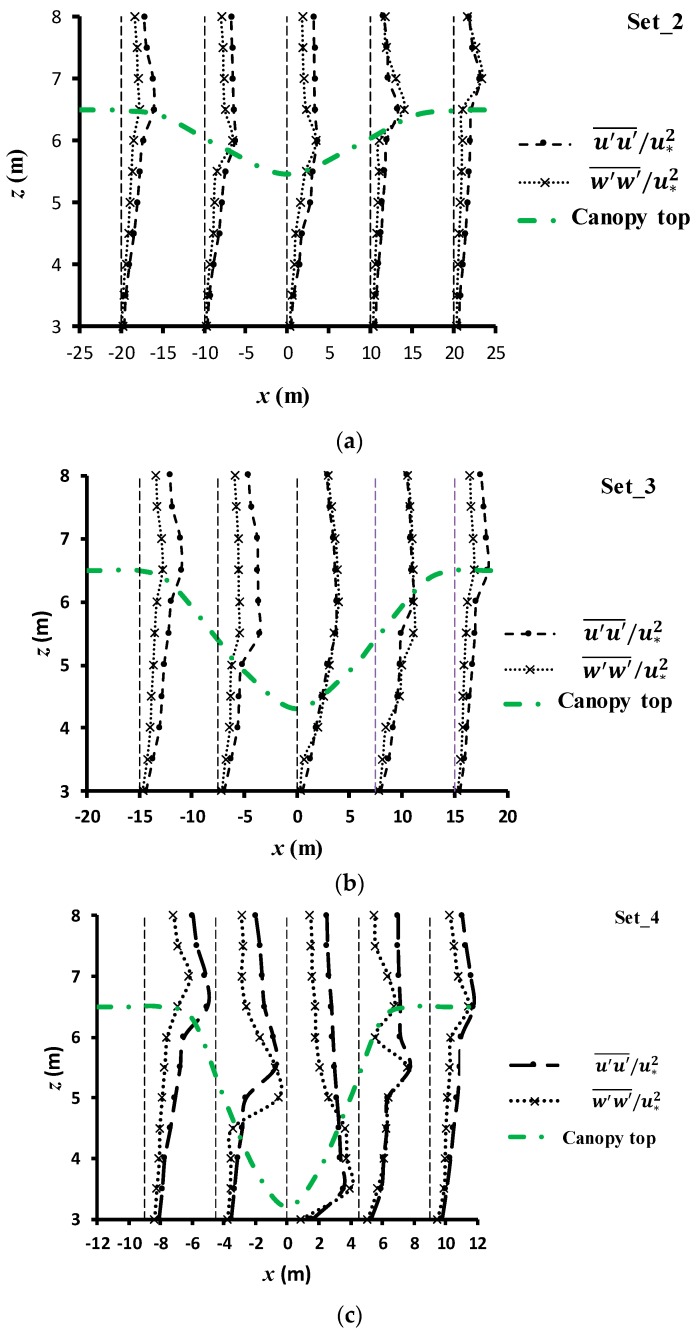
(**a**) Profiles of $\bar{u\prime u\prime }/u_{*}^{2}$ and $\bar{w\prime w\prime }/u_{*}^{2}$ for set_2 (**a**), set_3 (**b**), set_4 (**c**).

**Figure 13 ijerph-16-04846-f013:**
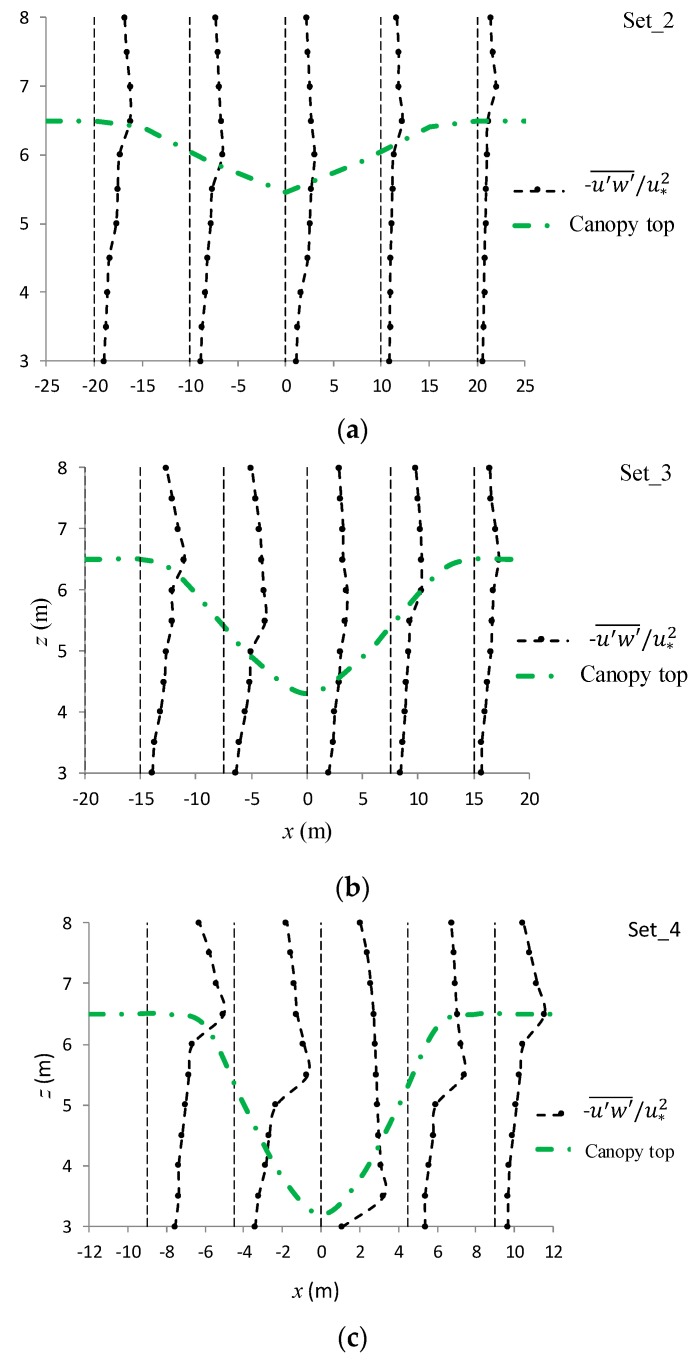
(**a**) Profiles of $-\bar{u\prime w\prime }/u_{*}^{2}$ for set_2 (**a**), set_3 (**b**), set_4 (**c**).

**Table 1 ijerph-16-04846-t001:** Parameters of the canopy caves.

Experiment No.	Depression Length *L* (cm)	Depression Depth *d* (cm)	Aspect Ratio *L*/*d*
Set_1	-	-	Flat canopy top
Set_2	36	1.0	36
Set_3	30	2.0	15
Set_4	24	3.0	8

**Table 2 ijerph-16-04846-t002:** Amount of captured solid pollutants in the canopy caves.

Experiment No.	Aspect Ratio *L*/*d*	Captured Solid Pollutants (g)	Enhance Rate of Foliage Capture (Times)
Set_1	Flat canopy top	0.65	1
Set_2	36	3.12	4.8
Set_3	15	5.39	8.3
Set_4	8	8.56	13.2

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
