# Peer review of "Bio-Capture of Solid Pollutants by Vegetation Canopy Cave in Shallow Water Flow"

_ijerph, 2019, doi:10.3390/ijerph16234846_

Round 1
Reviewer 1 Report
General Comments:
This paper investigated the "Bio-capture of solid pollutants by vegetation-canopy cave in shallow water flow".
It needs some major revisions.
A) Title should be write as "The potential of vegetation-canopy cave in bio-capture of solid pollutants (OR pollutant particles) in a shallow water flow".
B) English needs improvements.
C) There is no discussion!!! Authors need to compare their achievements with previous studies in details! Authors need to justify their results!
1. Abstract:
1.1. Page 1, Line 11; "The fine pollutant solids in..." Should be edited as "The fine solid pollutants in..." Or "The fine pollutant particles"
1.2. Page 1, Line 12; "In addition, they can infiltrate into soil and then into crops..." Should be edited as "In addition, they can infiltrate into soil and then accumulate in crops..."
1.3. Page 1, Line 13; "This paper aims to..." Should be edited as "This paper aimed to..."
1.4. Page 1, Line 18; "The results show that..." Should be edited as "The results revealed that..."
1.5. Write keywords alphabetically!
2. Introduction:
The introduction was written as a report!! You need to bring up gap of the knowledge and problem statement with using previous studies! Why did you conduct this research? What is the novelty of this study?
3. Materials and Methods:
Adequate, but needs some supports via previous studies.
4. Results and Discussion:
4.1. Authors just presented the results without discussing them!
4.2. Quality of Figures 9 and 12 needs improvement!
Author Response
Thanks for your valuable comments and suggestions.
Please see the attachment for the point-by-point response, thanks!

Reviewer 2 Report
This paper presents a physical experiment to investigate the influence of canopy-cave on pollutant’s captive performance. Authors used a “without-cave” configuration to compare the effect of aspect ratio of canopy-cave on the total amount of pollutant solids captured.
Authors need to modify the abstract section to include the unique features of the physical experiment and the main outcomes (i.e percent performance improvement). The paper has important message, and should be of great interest to the readers. However, authors could strength the paper by adding more discussion on the interpretation of result and comparison (i.e Section 3:1 and 3.2 could be strengthened by adding quantitative comparisons) It would also be interesting to see if the shape of vegetation foliage has significant effect in the pollutant solids captured. Authors need to include discuss on how the staggered arrangement of vegetation configuration is used. Line 180-181: provide more clarification for the statement “… near canopy-top the concentrations of fine pollutant solid are greater than its under and above regions” Line 190 -194: the outcomes contradict with the main conclusion. The total amount of captured pollutant increased by 73% from set_2 to set_3 with the decrease in aspect ratio, and then increases ONLY by 59% from set_3 to set_4. Is there is an optimal aspect ratio of canopy-cave? This contradict with the statement in Line 190 “…with the decrease of aspect ratio of canopy-cave, the total amount of captured pollutant solids is increasing” Conclusions are too general. These include: Results show that the aquatic vegetation canopy-cave can be applied as an effective bio technique to capture fine pollutant. Compared with flat canopy-top topography, the with-cave canopy improves the capture performance and capture rate significantly.Author Response
Thanks for your valuable comments and suggestions.
Please see the attachment for the point-by-point response, thanks!

Reviewer 3 Report
The authors present an interesting experimental study on solid contamination transport over the canopy field. I would suggest some changes to the manuscript before publication.
Even though the experiments are presented with care and the manuscript mostly reads well but there are some shortcomings that detract from overall content.
My main issue which I want to mention is related to velocity measurements and namely the turbulence and vorticity. In section 2.5 authors discuss the turbulence intensities for 2D flow in x-z plane. As to the longitudinal velocities I will not expect any issues making the calculations doubtful, the vertical velocities may rise some problems. First of all the clear definition of the turbulence intensity is needed as for instance shown in here:
Kolerski, T. and Wielgat, P., 2014. Velocity Field Characteristics at the Inlet to a Pipe Culvert. Archives of Hydro-Engineering and Environmental Mechanics, 61(3-4), pp.127-140.
I expect in some locations the vertical velocity will oscillate around zero, changing form positive to negative direction (up and down direction). Thus the method of averaging the vertical velocity will have a large impact on the archived results.
I am also not clear how the vorticity is calculated from the data which are measurement data, discrete in essence. Therefore solution of the equation (1) is not a trivial issue. Please provide solution method for equation (1) with applied numerical scheme.
Also, I think the title of the chapter ‘2.5. Two parameters’ should be changes to some better suited
Author Response

(The authors gave the same response as above.)

Round 2
Reviewer 1 Report
It seems that the reviewer's comments have been addressed well.
I recommend that the authors merge sections 3 and 4 together, 3. Results and Discussion
Reviewer 3 Report
Authors’ replays are satisfying. No additional questions.
In my opinion the paper can be published in present form